# What’s in a Meow? A Study on Human Classification and Interpretation of Domestic Cat Vocalizations

**DOI:** 10.3390/ani10122390

**Published:** 2020-12-14

**Authors:** Emanuela Prato-Previde, Simona Cannas, Clara Palestrini, Sara Ingraffia, Monica Battini, Luca Andrea Ludovico, Stavros Ntalampiras, Giorgio Presti, Silvana Mattiello

**Affiliations:** 1Department of Pathophysiology and Transplantation, University of Milan, 20133 Milan, Italy; 2Department of Veterinary Medicine, University of Milan, 20133 Milan, Italy; simona.cannas@unimi.it (S.C.); clara.palestrini@unimi.it (C.P.); sara.ingraffia94@gmail.com (S.I.); 3Department of Agricultural and Environmental Science, University of Milan, 20133 Milan, Italy; monica.battini@unimi.it (M.B.); silvana.mattiello@unimi.it (S.M.); 4Department of Computer Science, University of Milan, 20133 Milan, Italy; luca.ludovico@unimi.it (L.A.L.); stavros.ntalampiras@unimi.it (S.N.); giorgio.presti@unimi.it (G.P.)

**Keywords:** domestic cat, *Felis catus*, cat–human communication, meow, empathy, questionnaire

## Abstract

**Simple Summary:**

Cat–human communication is a core aspect of cat–human relationships and has an impact on domestic cats’ welfare. Meows are the most common human-directed vocalizations and are used in different everyday contexts to convey emotional states. This work investigates adult humans’ capacity to recognize meows emitted by cats during waiting for food, isolation, and brushing. We also assessed whether participants’ gender and level of empathy toward animals in general, and toward cats in particular, positively affect the recognition of cat meows. Participants were asked to complete an online questionnaire designed to assess their knowledge of cats and to evaluate their empathy toward animals. In addition, they listened to cat meows recorded in different situations and tried to identify the context in which they were emitted and their emotional valence. Overall, we found that, although meowing is mainly a human-directed vocalization and should represent a useful tool for cats to communicate emotional states to their owners, humans are not good at extracting precise information from cats’ vocalizations and show a limited capacity of discrimination based mainly on their experience with cats and influenced by gender and empathy toward them.

**Abstract:**

Although the domestic cat (*Felis catus*) is probably the most widespread companion animal in the world and interacts in a complex and multifaceted way with humans, the human–cat relationship and reciprocal communication have received far less attention compared, for example, to the human–dog relationship. Only a limited number of studies have considered what people understand of cats’ human-directed vocal signals during daily cat–owner interactions. The aim of the current study was to investigate to what extent adult humans recognize cat vocalizations, namely meows, emitted in three different contexts: waiting for food, isolation, and brushing. A second aim was to evaluate whether the level of human empathy toward animals and cats and the participant’s gender would positively influence the recognition of cat vocalizations. Finally, some insights on which acoustic features are relevant for the main investigation are provided as a serendipitous result. Two hundred twenty-five adult participants were asked to complete an online questionnaire designed to assess their knowledge of cats and to evaluate their empathy toward animals (Animal Empathy Scale). In addition, participants had to listen to six cat meows recorded in three different contexts and specify the context in which they were emitted and their emotional valence. Less than half of the participants were able to associate cats’ vocalizations with the correct context in which they were emitted; the best recognized meow was that emitted while waiting for food. Female participants and cat owners showed a higher ability to correctly classify the vocalizations emitted by cats during brushing and isolation. A high level of empathy toward cats was significantly associated with a better recognition of meows emitted during isolation. Regarding the emotional valence of meows, it emerged that cat vocalizations emitted during isolation are perceived by people as the most negative, whereas those emitted during brushing are perceived as most positive. Overall, it emerged that, although meowing is mainly a human-directed vocalization and in principle represents a useful tool for cats to communicate emotional states to their owners, humans are not particularly able to extract precise information from cats’ vocalizations and show a limited capacity of discrimination based mainly on their experience with cats and influenced by empathy toward them.

## 1. Introduction

Over the past two decades, scientific interest in the human–animal relationship and interaction has rapidly grown, leading to a large body of literature on both theoretical and practical aspects of this interspecific relationship. In particular, studies on domestic species have increased considerably, providing insight into the physiological, ethological, psychological, and sociocultural aspects of the multifaceted relationship between humans and nonhuman species [1,2,3,4,5,6].

Domestic species are considered interesting models for investigating interspecific relationships and communication since domestication, artificial selection, and close coexistence with humans have shaped their behavior and sociocognitive abilities, favoring the emergence of interspecific relationships based on mutual understanding, effective communication, and emotional connection (e.g., [7,8,9]). Even though the dog (*Canis familiaris*) is regarded as the archetype of a “companion animal” among the domestic species due to its unique sociocognitive and communicative abilities [3], a growing number of studies show that other domestic animals also have sociocommunicative abilities that allow them to interact and communicate with humans. For example, like dogs, domestic cats (*Felis catus*) [10,11], horses (*Equus caballus*) [12,13], goats (*Capra hircus*) [14], pigs (*Sus scrofa domesticus*) [15,16], and ferrets (*Mustela furo*) [17] are sensitive and respond to some humans’ communicative cues. In addition, cats [11], pigs [16], and horses [18,19] use communicative cues to manipulate the attention/behavior of a human recipient to obtain an unreachable resource. Last but not least, there is evidence that cats [20,21,22], horses [23,24], and goats [25] are able to recognize and respond to human emotional expressions.

Dogs and cats are the two most common nonhuman animals with which we interact. They have a long history of domestication and close association with humans [26,27], are beloved companion animals living in the human household, and are widely viewed as important social partners by their owners [28]. In some countries, cats are rapidly becoming extremely popular domestic animals not just for practical reasons, but also thanks to their flexibility in adapting to human environments and to their capacity to communicate in a complex way with humans, forming well-established relationships with them [29,30,31]. Differently from their wild ancestors (*Felis silvestris*), domestic cats are often defined to be social [32,33], as they show certain social interactions in particular circumstances (for example, around an abundant food source), and have sociocognitive and communicative abilities probably developed to maintain social groups [34,35] and to manage different social interactions with humans as well as other pets [2,36,37].

Although research on domestic cat behavior and cognition is growing [37], cat cognitive and communicative skills have been far less investigated than those of dogs, and the literature on the cat–human relationship and communication is more sparse and limited [10,38]. Only a handful of studies have investigated cat vocalizations and the characteristics of cat–human communication [39,40,41,42,43] and little is known about the human ability to recognize and classify the context and the possible emotional content of cat-to-human vocalizations [44,45,46]. However, understanding the extent of the effectiveness of the reciprocal communication between cats and humans is not only theoretically interesting but also relevant for cat welfare, since cats, like dogs, live in close contact with their human social partners and depend on them for health, care, and affection.

Domestic cats have a wide and complex vocal repertoire; it includes several different vocalizations that are emitted in different contexts and carry information about internal states and emotions, allowing pet cats to communicate with humans [2,42]. Among cat vocalizations, meows appear to be highly modulated by the context of emission, with meows produced in positive contexts differing in their pitch, duration, and melody from meows produced in negative contexts [41,42,47]. Meow vocalizations are particularly interesting for a number of reasons:they appear to be rare in cat-to-cat interaction and in cat colonies [40], but they are typical of cat–human interactions [34,35,42];undomesticated felids rarely meow to humans in adulthood [48];meows emitted by feral cats and by cats raised in the human household show differences in their acoustic parameters [40], suggesting they are shaped by the close relationship with humans;it has been suggested that the meow could be a product of domestication and socialization with humans, with a less relevant role in intraspecific communication [49,50].

Despite the fact that various studies showed that humans can correctly classify the vocalizations of different species according to their context of emission and emotional content (e.g., chimpanzees [51], pigs [52], dogs [53], and cats [44]), meows have been largely overlooked, and the few available studies have produced contrasting results [41,44,46,54].

Therefore, the main aim of the current study was to further investigate to what extent humans recognize meows emitted in three different familiar contexts (i.e., waiting for food, isolation, and brushing) that elicit different behaviors [55] and are supposed to trigger different emotional states (positive or negative).

Two previous studies [44,46] found that human classification of context-specific domestic cat meow vocalizations seems to be relatively poor. In particular, Nicastro and Owren [44] reported that humans’ classification accuracy of unfamiliar cat meows was just modestly above chance and that experience with cats had a positive effect on context classification of single calls. In addition, they reported a slight positive effect of experience and affinity for cats on the classification of the affective valence. Conversely, Ellis et al. [46] found that the classification of cat meows in different contexts was above chance only when the vocalizing cat was the owned cat and not when the vocalizations belonged to an unfamiliar cat. In another study, Schötz and van de Weijer [41] found that human listeners’ ability to classify cat meows recorded either during feeding time and while waiting at a vet clinic was significantly above chance, and listeners with cat experience performed significantly better than naive listeners. Finally, Belin et al. [54] reported that humans failed to recognize the emotional valence of cat meows recorded in affective contexts of positive or negative emotional valence (food-related and affiliative vs. agonistic and distress contexts).

The role of experience emerges in a number of studies (cats: [45]; pigs: [52]; different species: [51]). McComb and colleagues [45] found that participants with no experience of owning cats judged purrs emitted while cats were actively seeking food (i.e., solicitation purrs) as more urgent and less pleasant than those emitted in other contexts (nonsolicitation purrs). However, individuals who had owned a cat performed significantly better than nonowners. Similarly, Tallet and colleagues [52] reported that people with no experience of pigs were able to classify the context and detect the emotional content of piglet vocalizations: however, ethologists and farmers were more skilled in discriminating different emotions than naive people and the type of experience influenced the judgment of the emotional intensity of piglets’ vocalizations. Finally, Scheumann and colleagues [51] found that, in order to recognize emotions from humans’, chimps’, dogs’ and tree shrews’ vocalizations, human listeners had to be familiar not only with the species but also with the specific sound evoked by a given context. Thus, the second purpose of the present study was to further explore the effect of experience on the identification of cats’ meows context, and also to evaluate the potential role of empathy and gender.

In a psychological perspective, empathy refers to the ability to perceive, understand and share another individual’s emotional state [56,57], whereas from an evolutionary perspective empathy’s purposes are to promote prosocial, cooperative behavior and to understand or predict the behavior of others [58]. Studies on humans show that the ability to recognize the emotional states of others from vocal and/or visual cues appears to be positively influenced by both empathy and gender. More empathic individuals are more accurate in recognizing others’ emotional states [59,60,61]. Moreover, there is evidence that women are more empathic [62] and more skilled in recognizing emotions than men [63,64,65]. In the field of human-animal interaction the link between empathy toward animals and the capacity to recognize other species emotional states from visual/vocal cues has been poorly investigated so far [66,67]. Furthermore, although gender differences have been reported in many studies of human–animal interactions (e.g., [4]), very little is known on potential gender differences on the capacity to recognize other species’ emotional states [44,52]. Empathy toward animals seems to be a good predictor of how dog-owners and vets rate pain in dogs [66] and cattle [67]. In their study on human ability to recognize piglets emotional vocalizations, Tallet et al. [52] hypothesized that ethologists performed better then farmers and naive people in identifying the context in which piglets vocalizations were emitted and assigned them a more negative valence than did farmers, because they were more empathic since they are usually interested in animal welfare. They also reported a small gender difference in the evaluation of piglets’ vocalizations. Similarly, Nicastro and Owren [44] found a significant effect of “affinity” for cats in general on the classification of production context of single calls, and a gender difference that approached but did not reach statistical significance.

In the current study, adult human participants were asked to listen to audio recordings of single meows recorded in the home environment from 10 cats of the same breed (Maine Coon) belonging to a single private owner and thus sharing similar environmental conditions. Participants were asked to complete an online questionnaire assessing their knowledge of cats and their empathy toward animals and cats. They were also asked to listen to cat meows recorded in different familiar contexts and to specify the context in which the vocalizations were emitted and their emotional valence. Based on the available literature, we hypothesized that human participants should be able to classify meows recorded in the different behavioral contexts; we also expected them to recognize, at least to some extent, the emotional valence of the meowing cat (positive vs. negative). Moreover, we hypothesized that experience with cats, empathy toward animals and/or cats, and gender would facilitate participants’ performance in the classification task.

## 2. Materials and Methods

### 2.1. Participants

Two hundred and twenty-five adults (79 men and 146 women) ranging in age between 18 and 70 (40.47±15.50) participated in the study. Participants were recruited through personal contacts, word of mouth and by advertising the study on the researchers’ Facebook sites; therefore, all participants were volunteers with different levels of experience with pets and with cats in particular. Potential participants were told that “the purpose of the study was to investigate humans’ understanding of cats’ vocalizations and that they would have to fill in an online questionnaire and to listen to a number of cats’ vocalizations”.

### 2.2. Cat Vocalizations

In the context of an interdisciplinary project involving the departments of Pathophysiology and Transplantation, Veterinary Medicine, Agricultural and Environmental Science, and Computer Science of the University of Milan, a dataset called “CatMeows” was created. Such a dataset, publicly available on Zenodo [68] and described in detail in [69], is composed by sounds obtained from two cat breeds: European Shorthair and Maine Coon.

In the present work, we focused only on the 10 Maine Coon cats: meows were recorded from six females and four males, ranging in age between 1 and 13 years (5.9±3.3). Three males and three females were neutered. All cats belonged to a single private owner, and thus shared similar environmental conditions. All cats were fed *ad libitum* with “Royal Canine Sensible” dry cat food and twice a day with “Cosma Nature” canned cat food (at about 7:00 a.m. and 7:00 p.m.). All cats were brushed monthly since kittenhood in order to maintain healthy fur conditions. All the subjects were used to the pet carrier since kittenhood and entered it spontaneously when it was open. They were also used to being transported outside the home by car (inside the pet carrier) about once a year to go to the vet or during holidays. Each cat was exposed for five minutes in a random order to three different situations that normally occur in the life of a cat, in order to stimulate the production of meows in different contexts. The exposure to each experimental context was repeated three times for each cat, at a one-month interval. Meows were recorded in the following three contexts:Waiting for food (meows made prior to regular feeding)—The owner started the normal routine operations that precede food delivery in the home environment, but food was actually delivered with a delay of five minutes;Isolation (meows made during a period of isolation in an unfamiliar environment)—The cat was placed in its pet carrier and transported by its owner, adopting the same routine used to transport it for any other reason, to an unfamiliar environment (e.g., a room in a different apartment or an office, not far from their home environment). On arrival, the owner opened the pet carrier and the cat was free to roam in the room (if it wanted) for 30 min, in the presence of the owner, to recover from transportation stress. Then, the cat was left alone in the room for five minutes;Brushing (meows made while being brushed by the owner)—Cats were brushed by their owner in their home environment for a maximum of 5 min.

Cat meows were recorded by using the small Bluetooth microphone of a *QCY Q26 Pro* Bluetooth headset placed on the cats’ collar. All cats were used to wearing the collar and had previously been accustomed to the presence of such a device (Figure 1). Recordings, saved as WAV files with a sampling frequency of 8000 Hz and 16 quantization bits, were subsequently subjected to bioacoustic analyses in order to select two meows for each situation: a medioid (i.e., the most representative sound of the specific situation) and an outlier (i.e., the most different sound with respect to the typical one emitted in the specific situation). Spectrograms of the six selected sounds are visible in Figure 2.

Medioids and outliers were identified by computing three audio features (see [70] for further information about audio features) for each recording, namely the fundamental frequency $F_{0}$, the roughness *R*, and the tristimulus $T_{1-3}$:$F_{0}$ corresponds to the pitch (or note) of the vocalization. It has been computed with the SWIPE’ method [71], ignoring values where the confidence reported by SWIPE’ is below 0.15;*R* is defined in [72] as the sensation for rapid amplitude variations, which reduces pleasantness, and whose effect is perceived as dissonant [73]. This is a sonic descriptor which conveys information about unpleasantness. It was computed using the dedicated function of the MIRToolbox [74] and, more specifically, adopting the strategy proposed in [75];$T_{1-3}$ is a set of features coming from color perception studies that has been adapted to the audio domain [76]. $T_{1}$ is the ratio between the energy of $F_{0}$ and the total energy, $T_{2}$ is the ratio between the second to fourth harmonics and the total energy, and $T_{3}$ is the ratio between all others harmonics and the total energy. Tristimulus was chosen to represent information regarding formants, since, with a band-limited signal containing only few harmonics, the actual computation of formants with linear prediction techniques resulted to be unreliable.

All the above-mentioned features are time-varying signals, extracted from the short-term Fourier transform of the sound. In order to reduce them to a set of global values, mean values and standard-deviation ranges were calculated for each feature. This led to a feature space of 10 values. Finally, a one-way ANOVA test was carried out to assess which features are meaningful in distinguishing the three situations (i.e., p<0.05), an operation that discarded $T_{2}$ and $T_{3}$, leading to a final feature space of 6 values per sound.

By taking the mean of the feature space of each class, three centroids were found. Medioids were chosen as the sounds closest to the centroids in terms of Euclidean distance, while outliers were the most distant from the centroids. Outliers were included mainly for two reasons: first, to have at least a pair of samples for each class, but keeping them sufficiently distant from each other (i.e., to sample the data space with some criterion), and second, to assess if the chosen features were effective in grasping the informative content of meows. In particular, we expect a significant difference in the recognition scores in favor of the medioids if the features are actually effective (and no significant differences otherwise).

### 2.3. Questionnaire

The whole questionnaire (Appendix A) included three sections. The first section contained general information on participants: age, gender, education, and other background experiences which could be relevant in determining their responses (previous and present interaction/experience with animals, past or actual pet ownership, and cat ownership).

The second section was aimed at evaluating the human–cat relationship and comprised the Animal Empathy Scale (AES), designed to measure empathy toward animals [77] and the Cat Empathy Scale (CES), calculated on the basis of three additional specific questions investigating participants’ reported empathy toward cats. The Animal Empathy Scale (AES) was initially developed by Paul [77] to measure empathy toward animals. It has been recently translated into Italian and used in two studies assessing empathy toward animals in a sample of students of veterinary medicine at the University of Milan (Italy) and in veterinary practitioners working with pets [78,79]. The scale includes a total of 22 items, 11 representing unempathic sentiments and 11 empathic sentiments. Responses to each item are requested on a 9-point Likert-type scale ranging from “very strongly agree” to “very strongly disagree”, with agreements with empathic statements scoring high (maximum 9) and agreements with unempathic statements scoring low (minimum 1). The total score, ranging from a minimum of 22 to a maximum of 198, is calculated as the sum of scores obtained in each item. Higher scores indicate a higher level of self-reported empathy. Previous studies carried out on different samples showed that the AES has a good internal consistency, with Cronbach alpha values ranging from 0.78 [77] to 0.83 [78] and 0.68 [79].

Since the AES evaluates empathy toward animals in general with only two items specifically referring to cats (item 2: Often cats will meow and pester for food even when they are not really hungry; item 9: A friendly purring cat almost always cheers me up), in order to specifically investigate the level of empathy toward cats (CES) reported by participants, the following three additional questions were added: 1. I can easily understand whether a cat is trying to communicate with me; 2. I can easily and intuitively understand how a cat is feeling; 3. I am good at predicting what a cat will do. As for the AES, responses to these questions were requested on a 9-point Likert-type scale ranging from “very strongly agree” to “very strongly disagree”. The total score ranged from a minimum of 3 to a maximum of 27 and was calculated as the sum of the scores for each question. Higher scores indicated a higher level of self-reported empathy toward cats.

In the third part of the survey, participants were asked to wear earphones and listen with attention to the recordings of six meows (one medioid and one outlier per context), specifically prepared for the study, as described above. After listening to each meow (participants could replay each sound ad libitum), participants were asked to choose one out of three possible contexts in which the vocalization was emitted (i.e., waiting for food, isolation, brushing), to indicate the emotional state of the meowing cat in terms of valence (positive vs. negative), and to give a score on a 7-point Likert scale to 11 descriptors of the possible emotional state: agitated/anxious, aggressive/angry, frustrated, restless/nervous, frightened, suffering, friendly, calm/relaxed, happy, curious, and playful. This procedure was based on previous research using qualitative behavior assessment in companion animals (dogs) [80] and reminds sociolinguistic researchers to investigate attitudes toward different aspects of language [81,82]. Between the proposed sounds, a 5-second pink noise sample was reproduced in order to avoid direct comparison.

### 2.4. Procedure

The questionnaire developed for the study was made available online from March 2018 until March 2019. All subjects who voluntarily accessed the questionnaire were told that the purpose of the survey was to gain knowledge regarding the human–animal relationship and that their responses would remain anonymous and be used for scientific research only. They also signed an informed consent form and an authorization to allow us to use the data according to the National Privacy Law 675/96. Participants were also asked to fill the entire questionnaire and to listen to the vocalizations in a quiet moment of the day, taking their time and using their earphones, and to carefully follow the guided procedure.

### 2.5. Ethical Statement

The present project was approved by the Animal Welfare Organisation of the University of Milan (approval n. OPBA_25_2017). The challenging situations were conceived considering potentially stressful situations that may occur in cats’ life and to which cats can usually easily adapt. In order to minimize possible stress reactions, preliminary information on the normal husbandry practices (e.g., brushing or transportation) to which the experimental cats were submitted and on their normal reactions to these practices were collected in interviews with the owners. The information collected did not point out any possibility of excessive reactions of cats in one of the planned situations; therefore, all the cats were included in the trial. No signs of excessive stress were ever recorded in any of the challenging situations, all of which could therefore be completed. Before starting to complete the questionnaire, the interviewed people were asked to sign an informed consent, stating that all data were going to be treated anonymously and used only for scientific purposes.

## 3. Statistical Analysis

All data were collected in a number of spreadsheets for statistical analyses. Preliminary descriptive analyses were carried out to evaluate the characteristics of the sample and the distribution of the data collected.

The accuracy rate of the responses given by the participants (percentage of correct assignment) was calculated for each context and compared between medioids and outliers using chisq test. Then, taking into consideration only medioids, the accuracy rate of the responses among contexts was compared using the chisq test. The chisq test was also used to compare, within each context, the accuracy rate of meow medioids depending on the following characteristics of the interviewed persons: gender (males vs. females), parental status (parents vs. nonparents), and level of experience with cats (cat owners vs. nonowners; grown up with cats vs. grown up without cats). The Mann–Whitney test was used to evaluate the effect of gender (males vs. females) and of the level of experience with cats (cat owners vs. nonowners) on AES and CES scores. The internal consistency of the AES was assessed using Cronbach’s alpha. Spearman correlations were calculated between AES and CES scores and the total number of correct context identifications (0 = no correct assignment; 1 = one correct assignment; 2 = two correct assignments; 3 = all assignments were correct). Within each context, AES and CES scores were compared, depending on the correct or incorrect assignment of the meow, using the Mann–Whitney test. Finally, in order to understand the type of emotions perceived by participants in response to the meows emitted in each specific context, we performed a principal component analysis (PCA) on the scores given to each descriptor. This analysis was initially performed on the whole sample, and then only using the descriptors of the situations that had been correctly assigned to their context. All the statistical analyses were carried out with SPSS Statistics 25 (IBM, Armonk, NY, USA), with alpha set at 0.05.

## 4. Results

In all contexts, medioid meows had a significantly higher probability to be assigned to the correct context than outlier meows (Table 1). This seems to prove that the chosen features are somehow related to those to which subjects are sensitive.

However, even considering only the medioids, the accuracy ratio of the responses was generally low, and it was never significantly above the chance level (0.33%). Statistical differences in the accuracy ratio of the responses were recorded among contexts (p<0.01): although still low and not above chance, waiting for food had the highest rate of correct assignment (40.44%), while isolation had the lowest (26.67%) (Table 1). Some individual characteristics of the interviewed persons significantly affected the accuracy rate (Table 2). Females showed a higher accuracy rate, with significant differences emerging in the isolation and brushing contexts, whereas parental status did not affect the accuracy rate. The level of experience with cats also had some effect on the accuracy rate: in particular, cat owners had a higher accuracy rate than nonowners in all contexts, with significant differences for isolation and brushing, whereas having grown up with cats did not affect the accuracy rate.

Both experience with cats and gender significantly affected the empathy toward animals in general (AES), and more specifically toward cats (CES), that were higher in females, cat owners, and persons who had grown up with cats (Table 3).

AES and CES scores were significantly correlated (σ = 0.316; p<0.001). A significant correlation was found between the total number of correct context identification and CES (σ = 0.145; p<0.05), whereas no correlation emerged between the total number of correct context identification and AES (σ = 0.040; n.s.). Within each context, AES and CES scores did not differ depending on the correct or incorrect assignment of the meow, except for CES in the isolation context (Table 4).

A preliminary PCA carried out on the scores given to each descriptor related to the type of emotion perceived by the interviewed participants to the meows emitted in each context revealed no clear separation among the emission contexts (data not presented). The same analysis conducted only on the descriptors of the meows that had been correctly assigned to their context showed a clear trend of meows emitted during brushing to cluster on the left side of PC1, whereas meows emitted in the isolation context are scattered on the right side and those emitted while waiting for food are in an intermediate position, with a trend to cluster on the left (Figure 3a).

The left side of PC1 is related to a positive valence, as shown by the high loadings of variables such as content, calm/relaxed, friendly, playful, and curious, whereas the right side is characterized by a negative valence, as shown by descriptors like nervous, frightened, agitated/anxious, frustrated, suffering, and angry/aggressive (Figure 3b). This means that brushing and, to a lesser extent, waiting for food are perceived by the interviewed persons as more positive situations, whereas isolation is clearly perceived as a negative situation.

The first two principal components explained 76.3% of total variance (PC1: 61.7%; PC2: 14.6%).

## 5. Discussion

The current study investigates the human–cat relationship and communication evaluating adult humans’ ability to classify single cat meows emitted in different well-defined and familiar contexts: waiting for food, brushing, and isolation. We also evaluated the effect of factors such as experience with cats, gender, and empathy toward animals and cats on human performance in the context recognition task. Finally, we asked participants to judge the emotional state of the meowing cat by scoring different descriptors, in order to highlight the perceived emotional valence (positive vs. negative).

Meowing is a common and mainly human-directed vocalization [34,35,42]; thus, in principle, it should represent a useful mean for cats to communicate their emotional states to humans. Furthermore, previous studies showed that meows emitted in these three different contexts can be successfully discriminated on the basis of a series of acoustic parameters [83]. In spite of this, our results suggest that adult humans have a limited capacity to discriminate among the production contexts of single meows (both outlier and medioid meows) in familiar contexts, and therefore seem unable to extract reliable specific information from cats’ meows.

In all contexts, the most representative meows of the specific situation (i.e., medioids) were assigned by participants to the correct context significantly more than meows with less core features, and thus more different from the typical sound emitted in the specific situation (outlier meows). However, even when taking into consideration exclusively meows most representative of a given context, the correct ratio of context assignment was never significantly above the random chance (0.33%). In particular, the best recognized meow was that emitted in the context “waiting for food”, with a rate of correct assignment of 40.44% (outlier meow = 27.11%), whereas the less recognized one was that emitted in the context “isolation”, with a rate of only 26.67% (outlier meow = 14.22%).

It is worth noting that the contexts in which cat meows were recorded, in particular waiting for food and isolation, were similar to the contexts used in previous studies [44,46,54] and were chosen because they are considered common ones for cats and humans and are associated to different emotional valence. However, differently from the results by Nicastro and Owren [44], our participants did not perform above chance in the context classification task. In this respect, our results are more in line with those by Ellis et al. [46], showing that vocalizations of unfamiliar cats are difficult to classify, and performance is above chance only when the vocalizing cat is the owned cat.

The finding that cat ownership positively affected classification is in line with previous work on cats [44,45,46] and other species [51,52], showing that experience with the species allows a better recognition of animal vocalizations. Despite the overall poor performance, cat owners were more accurate than nonowners in all contexts, with significant differences emerging for isolation and brushing, whereas having grown up with cats did not affect the accuracy rate. This suggests that regular and daily interactions may be more relevant than past experience in favoring the correct attribution. However, this seems to be true for acoustic cues, but not for visual cues: in fact, Dawson et al. [84] found that personal experience with cats (i.e., having ever lived with a cat, the number of years spent living with cats, the current number of owned cats) was not important for the correct identification of feline emotions from cats’ faces, and a previous study by Schirmer et al. [85] reported that humans without previous experience with dogs discriminate dogs’ facial expressions of both negative and positive emotions.

Current findings, together with previous ones, are in contrast with a series of playback experiments with dogs’ vocalizations showing that adult humans are successful in the recognition of contextual and motivational content of dog barks [53,86] independently from previous experiences, and that even children aged 6–10 years can perform similarly to adults [87].

It has been suggested that classifying the vocalizations of unfamiliar animals exclusively on the basis of mere acoustic cues, without the associated visual cues (e.g., facial, body expression), is a difficult task [44]; moreover, the variability of cat meows, both within and between contexts and individuals, may represent a further aspect that complicates the classification task in both experimental and natural circumstances. However, there is evidence that meows are highly modulated by the context of emission, with meows produced in positive contexts differing in their pitch, duration, and melody from meows produced in negative contexts [42,47], and that they can be automatically classified with a high accuracy rate on the basis of their acoustic characteristics [83]. Schötz et al. [47], for example, reported effects of the recording context and of cat internal state on f0 and duration of cat meows; in addition, they found that positive (e.g., affiliative) contexts and internal states tended to have rising f0 contours, while meows produced in negative (e.g., stressed) contexts and mental states had predominantly falling f0 contours. These acoustic characteristics could presumably help humans in differentiating at least negative versus positive emotional states. Interestingly, Yin [88] and Pongrácz et al. [53] reported that dog barking has reliable acoustic features (e.g., peak and fundamental frequency, interbark intervals) that are specific to particular contexts or inner states and that are used by humans to detect the context and the inner state of the barking dog. In the current study, the medioid meow was obtained by keeping the more representative aspects of the meow vocalization in a given context; thus, we expected humans to perform at least moderately above chance.

Regarding the effect of gender, our findings provide new insight on possible gender differences in human–animal interactions, suggesting a potential gender bias in the capacity to recognize other species emotional states from vocalizations. With regards to cats’ vocalizations, Nicastro and Owren [44] reported only a small gender effect that approached but did not reach significance; similarly, Tallet et al. [52] found small gender differences in the evaluation of piglet vocalizations and related them with the evidence of a greater empathy [89] and a greater accuracy in recognizing emotional expressions [65,90] in women than in men.

Studies on humans have shown that women have greater abilities to recognize nonverbal displays of other humans’ emotions [91], but little evidence is available with respect to other species. A recent study [84] showed that people can identify feline emotions from cats’ faces and reported that women were more successful than men in the identification of feline emotions. Similarly, Schirmer et al. [85] reported that women were more sensitive than men to dogs’ affective expressions.

These results are possibly due to the higher empathy of women toward animals recorded also in the present study, in line with our predictions and with the results of previous studies [62]. In fact, besides observing a correlation between empathy toward animals and cats, we also found that the total scores of empathy toward animals and cats were significantly higher in females. Evidence of females’ greater levels of empathy toward nonhuman animals has been reported in previous studies and is associated with more positive attitudes toward animals and a greater concern for their welfare [77,78,79].

Empathy was higher also in cat owners and persons who had grown up with cats, and a significant correlation was found between the total number of correct context identifications and empathy toward cats, but not empathy toward animals in general. Greater empathy toward cats, and not just toward animals in general, could motivate the choice of a cat as a companion animal; moreover, “being empathetic” toward cats may, for example, motivate owners to shift their attention toward them more, which can thereby increase accuracy in emotion recognition as happens in humans [60,61]. In general, feeling empathy together with familiarity can lead perceivers to focus on expressive cues that communicate information about the feelings of others. Of course, these are just hypotheses that require further investigation.

With regard to the evaluation of the emotional state of the meowing cat in terms of valence (positive vs. negative), the analysis carried out only on the descriptors of the meows that had been correctly assigned to their context showed that meows emitted during isolation were perceived by participants as more negative (the cat was described as nervous, frightened, agitated/anxious, frustrated, suffering, and/or angry/aggressive), whereas those emitted during brushing and, to a lesser extent, waiting for food were perceived as positive (the cat was described as calm/relaxed, friendly, playful, and/or curious). Since meows produced in positive contexts are different in their pitch, duration, and melody from meows produced in negative contexts [42,47], this finding suggests that the correct distinction of meows in terms of valence was probably based on the detection of these differences.

Nicastro and Owren [44], when grouping the different contexts by their presumptive affective valence, assigned the food-related context to the positive affect category, whereas calls made when the cats were placed in an unfamiliar environment (i.e., distress) were considered to be negative. However, unexpectedly, Ellis et al. [46] reported that meow vocalizations emitted while cats were alone in a room and unable to exit (i.e., negotiating a barrier context) were rated by participants as the most pleasant. Finally, Belin et al. [54] reported that humans failed to recognize the emotional valence of cat meows recorded in affective contexts of positive or negative emotional valence (food-related and affiliative vs. agonistic and distress).

The finding that meows emitted during isolation were perceived by people as negative whereas those emitted during brushing and, to a lower extent, waiting for food were perceived as positive supports previous evidence on cats and other species showing that humans are able to differentiate negative from positive emotions conveyed through vocalizations [44,53,54]. One possible reason why waiting for food was perceived as less positive than brushing in the current study could be that cats were gently brushed and touched by their owner as usual while in the brushing condition, whereas in the waiting for food context the owner, after starting the normal routine operations that preceded food delivery, waited for 5 min before delivering it, thus possibly inducing also some distress. In fact, as reported by Cannas et al. [55], cats waiting for food showed yawning, lip licking, swallowing, and salivation. These behaviors are related to a condition of stress and frustration due to the delay in giving food instead of its request. By definition, frustration arises when an individual is unable to immediately access something it wants [92].

## 6. Conclusions

Although cats’ popularity as pets rivals that of dogs, cats are far less studied than dogs, and cat–human communication has received less attention compared to dog–human communication. Our findings provide the first evidence of gender differences in the ability to recognize cats’ meow vocalizations and highlight the role of experience and empathy toward cats in the recognition of the context of emission of the vocalizations. However, overall, the current knowledge on human ability to decode cat meows indicates that, although meowing is a common and mainly human-directed vocalization and its acoustic characteristics vary depending on the context of emission, adult humans show a limited capacity to extract specific information from cats’ meows and poorly discriminate among the production contexts of single meows emitted in familiar contexts. Given the limited number of studies on cat-to-human vocalizations and the mixed results obtained so far, future studies should further explore human understanding of cat vocal communicative sounds and the different variables that affect it, as well as the human ability to understand the valence and arousal of emotional states elicited by different contexts.

## Figures and Tables

**Figure 1 animals-10-02390-f001:**
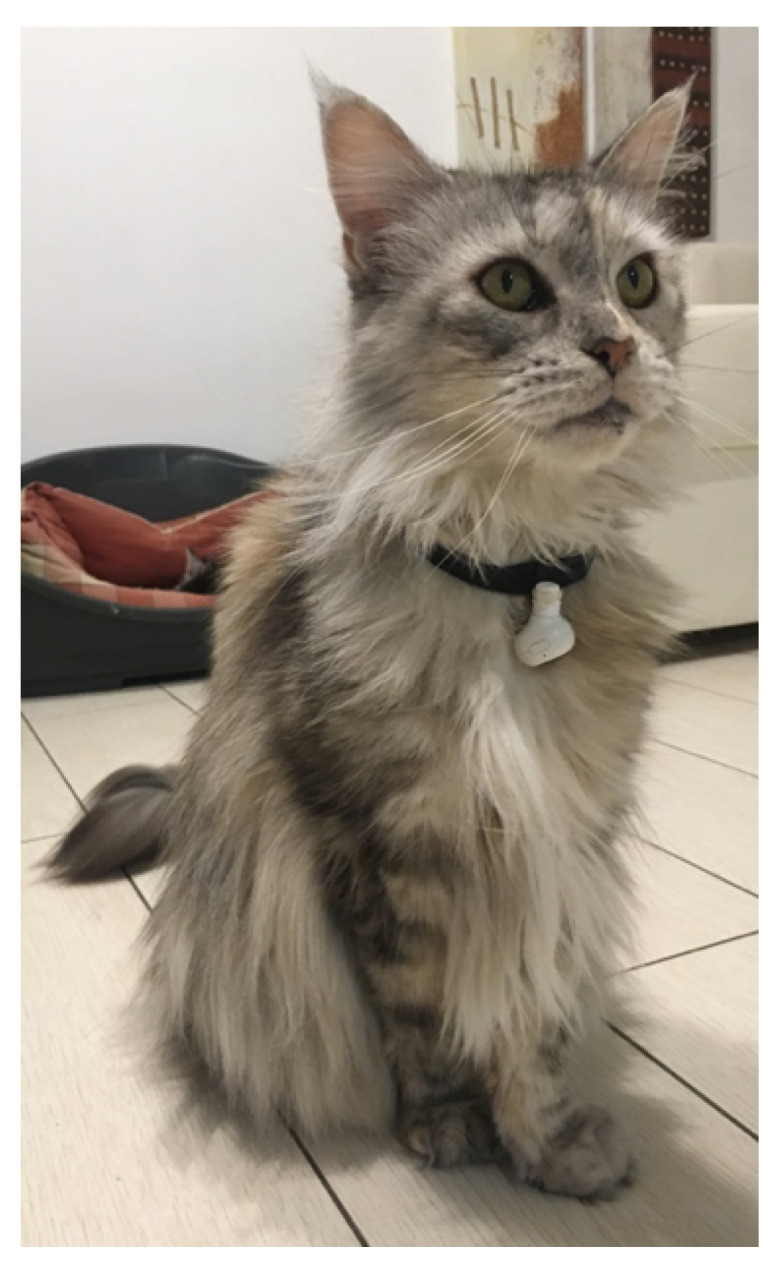
A cat provided with a Bluetooth microphone placed on the collar.

**Figure 2 animals-10-02390-f002:**
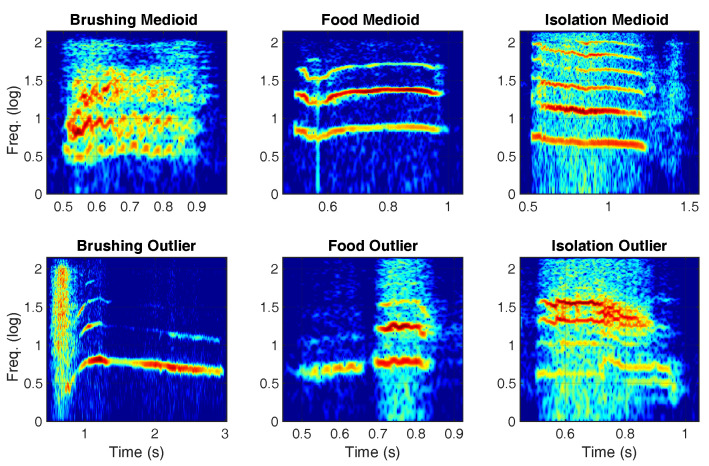
Time–frequency spectrograms of the medioid and outlier meows in the three different contexts.

**Figure 3 animals-10-02390-f003:**
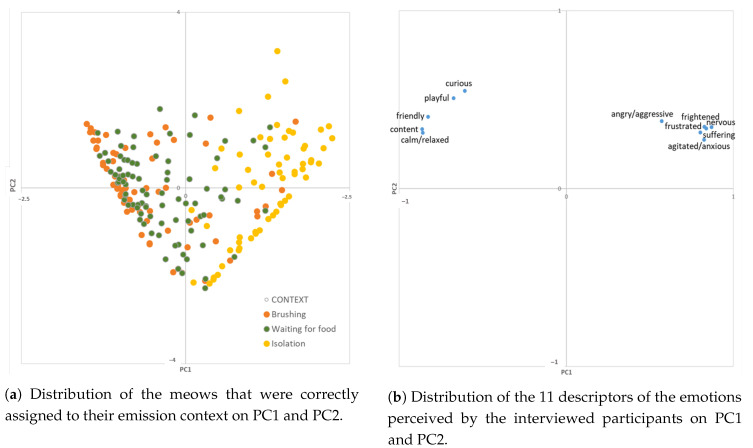
Two-dimensional score plot of PCA results.

**Table 1 animals-10-02390-t001:** Absolute frequencies (and percentages) of correct or incorrect assignment of medioid or outlier meows in each of the three contexts. Significance levels refer to differences between the rate of correct assignment of medioid vs. outlier meows within each context.

Context	Medioid Assignment		Outlier Assignment		Significance
	Correct	Incorrect	Correct	Incorrect	
Waiting for food	91 (40.44%)	134 (59.56%)	61 (27.11%)	164 (72.89%)	*p* < 0.01
Isolation	60 (26.67%)	165 (73.33%)	32 (14.22%)	193 (85.78%)	*p* < 0.001
Brushing	74 (32.89%)	151 (67.11%)	30 (13.33%)	195 (86.67%)	*p* < 0.001

**Table 2 animals-10-02390-t002:** Absolute frequencies (and percentages) of correct or incorrect assignment of medioid meows in each of the three contexts, depending on individual characteristics and level of experience with cats of the interviewed persons.

	Waiting for Food			Isolation			Brushing		
	Correct	Incorrect	Sign.	Correct	Incorrect	Sign.	Correct	Incorrect	Sign.
Gender									
Male	27 (34.2%)	52 (65.8%)	n.s.	14 (17.7%)	65 (82.3%)	p<0.05	19 (24.1%)	60 (75.9%)	p<0.05
Female	64 (43.8%)	82 (56.2%)		46 (31.5%)	100 (68.5%)		55 (37.7%)	91 (62.3%)	
Parental status									
Parent	24 (33.3%)	48 (66.7%)	n.s.	14 (19.4%)	58 (80.6%)	n.s.	18 (25.0%)	54 (75.0%)	n.s.
Nonparent	67 (43.8%)	86 (56.2%)		46 (30.1%)	107 (69.9%)		56 (36.6%)	97 (63.4%)	
Cat owner									
Yes	48 (44.4%)	60 (55.6%)	n.s.	38 (35.2%)	70 (64.8%)	p<0.01	48 (44.4%)	60 (55.6%)	p<0.001
No	43 (36.8%)	74 (63.2%)		22 (18.8%)	95 (81.2%)		26 (22.2%)	91 (77.8%)		
Grown up with cats									
Yes	51 (41.8%)	71 (58.2%)	n.s.	38 (31.1%)	84 (68.9%)	n.s.	44 (36.1%)	78 (63.9%)	n.s.
No	40 (38.8%)	63 (61.2%)		22 (21.4%)	81 (78.6%)		30 (29.1%)	73 (70.9%)	

**Table 3 animals-10-02390-t003:** Mean (±SD) of AES and CES of the participants, depending on their gender or experience with cats, and relative significance levels.

	AES		CES	
	Mean ± s.d.	Sign.	Mean ± s.d.	Sign.
Gender				
Male	144.28 ± 17.95	*p* < 0.001	15.92 ± 6.59	*p* < 0.05
Female	157.82 ± 18.02		17.97 ± 5.68	
Parental status				
Parent	153.58 ± 21.08	n.s.	18.32 ± 5.64	n.s.
Nonparent	152.82 ± 18.14		16.75 ± 6.23	
Cat owner				
Yes	157.11 ± 17.58	*p* < 0.01	20.06 ± 4.53	*p* < 0.001
No	149.32 ± 19.73		14.65 ± 6.18	
Grown up with cats				
Yes	157.75 ± 17.20	*p* < 0.001	19.03 ± 5.08	*p* < 0.001
No	147.50 ± 19.79		15.14 ± 6.50	
Overall	153.06 ± 19.09		17.25 ± 6.08	

**Table 4 animals-10-02390-t004:** Mean (±SD) of AES and CES of the participants depending on the correct or incorrect assignment of meows to each of the three contexts.

	AES			CES		
Context	Context Assignment			Context Assignment		
	Correct	Incorrect	Sign.	Correct	Incorrect	Sign.
Waiting for food	154.02 ± 18.32	152.41 ± 19.63	n.s.	17.90 ± 5.55	16.81 ± 6.40	n.s.
Isolation	154.92 ± 16.48	152.39 ± 19.95	n.s.	18.75 ± 5.38	16.70 ± 6.24	*p* < 0.05
Brushing	154.01 ± 18.92	152.60 ± 19.21	n.s.	18.31 ± 5.65	16.73 ± 6.23	n.s.

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
