# Peer review of "What’s in a Meow? A Study on Human Classification and Interpretation of Domestic Cat Vocalizations"

_animals, 2020, doi:10.3390/ani10122390_

Round 1

Reviewer 1 Report

Review on the manuscript (animals-998464) “What’s in a Meow? A Study on Human Classification and Interpretation of Domestic Cat Vocalizations”

Written by Prato-Previde et al.

Overall comments

I read this manuscript mostly with great pleasure as (1) it has a very interesting and relevant topic about cat-human communication; (2) it is written in an extraordinarily high quality. I found both the Introduction and the Discussion excellent, modest and comprehensive; the experimental design as being intuitive and careful.

All in all, it would be an almost perfect study, with one major, and for me at least, hardly explainable flaw. This is the extremely low number of sound samples the Authors provided to the participants to evaluate. There were three social contexts where cat meows were originally collected from, and altogether 10 cats provided sound samples. Unfortunately, Authors used only 2 meows from each context for the playback study. The two meows were selected in each context after a bioacoustical analysis, which enabled the authors to pick a typical (‘medioid’) and an atypical (‘outlier’) exemplar meow for each context. Although I fancy this method of selection, I still must say that using this few sound samples (realistically, ONE (!) sample per context) is way below the acceptable level of representative sampling. In vain more than two hundred human listeners evaluated the sounds, the pseudo-replication effect is enormous here. In a nutshell, for example measuring 100 times the same three objects (one from each of the three categories) is not the same than measuring 10 times 10 objects from 3 categories. I really like this study and all the comparisons and analyses Authors performed within, but I simply cannot endorse a method where only one (context-typical) sound sample was used from each situation.

Detailed comments

Line 52 - I recommend using “forming well-established relationship with them” instead of “forming strict relationship with them”

Regarding the Results, I would recommend to analyze whether the correctly recognized proportions of meows in each context would exceed significantly the chance level (that is 33.33% in case of three possible choice options). Using the 50% as a reference level is not correct, because there were three context-options to choose from.

Author Response

Response to Reviewer 1 Comments

Point 1: All in all, it would be an almost perfect study, with one major, and for me at least, hardly explainable flaw. This is the extremely low number of sound samples the Authors provided to the participants to evaluate. There were three social contexts where cat meows were originally collected from, and altogether 10 cats provided sound samples. Unfortunately, Authors used only 2 meows from each context for the playback study. The two meows were selected in each context after a bioacoustical analysis, which enabled the authors to pick a typical (‘medioid’) and an atypical (‘outlier’) exemplar meow for each context. Although I fancy this method of selection, I still must say that using this few sound samples (realistically, ONE (!) sample per context) is way below the acceptable level of representative sampling. In vain more than two hundred human listeners evaluated the sounds, the pseudo-replication effect is enormous here. In a nutshell, for example measuring 100 times the same three objects (one from each of the three categories) is not the same than measuring 10 times 10 objects from 3 categories. I really like this study and all the comparisons and analyses Authors performed within, but I simply cannot endorse a method where only one (context-typical) sound sample was used from each situation.

Response 1: We chose two sounds emitted by two cats for each class: one "typical" (the medioid, ideal in case we had to choose only one sound) and the other sufficiently different from the first one (so as to maximize the coverage of the meow space and have a backup in case the biometrics were not representative). Having 10 cats in total, this means a coverage of 20% of the cats, a percentage reasonable for our goals. Anyway, we are planning to extend it in our future works, when we will also have a richer dataset

Point 2: Line 52 - I recommend using “forming well-established relationship with them” instead of

“forming strict relationship with them”

Response 2: Done (Line 52)

Point 3: Regarding the Results, I would recommend to analyze whether the correctly recognized proportions of meows in each context would exceed significantly the chance level (that is 33.33% in case of three possible choice options). Using the 50% as a reference level is not correct, because there were three context-options to choose from.

Response 3: We thank the reviewer for this useful comment. Actually, we never used 50% as a reference level. In Table 1 we just present a comparison between the correct classification rate of medioids compared with the correct classification rate of outliers. However, the comment was quite useful, and we have included a comparison of the correct classification rate of medioids with the chance level (n.s. for all contexts), as suggested by the reviewer. We therefore reformulated the original in the Results section as follows: “However, even considering only the medioids, the accuracy ratio of the responses was generally low, and it was never significantly above the chance level (0.33%). Statistical differences in the accuracy ratio of the responses were recorded among contexts (p<0.01): although still low and not above chance, waiting for food had the highest rate of correct assignment (40.44%), while isolation had the lowest (26.67%) (Table 1)(L 295-299 in the new version). Additionally, we modified a sentence the Discussion, which now reads: “However, even when taking into consideration exclusively meows most representative of a given context, the correct ratio of context assignment was never significantly above the random chance (0.33%). (L 344-346 in the new version).

Reviewer 2 Report

A well-written, engaging paper, publishable as is. Just two minor comments:

1) just above section 2.4, the sentence that reads as follows:

After listening to each meow (participants could replay each sound ad libitum ) participants were asked to indicate the specific situation in which the vocalization was emitted (i.e. waiting for food, isolation, brushing), to indicate the emotional state of the meowing cat in terms of valence (positive vs. negative)...

It isn't entirely clear that the participants were given only those three alternatives (waiting for food, isolation, brushing) or whether they were free to offer other suggestions. I believe that those three alternatives were the only ones available to choose from, but because the sentence as it is written is ambiguous, I encourage a small revision/clarification there. 

2) in the discipline of sociolinguistics there is something called a "matched-guise test", which operates in a very similar way to the experimental technique described here. Because an important point of this paper is the investigation of attitudes towards the cat vocalizations that participants heard, I encourage the authors to include one or two references to that literature; if for no other reason to signal that the text is accessible to and may perhaps be useful to sociolinguists interested in human-animal communication. 

Author Response

Response to Reviewer 2 Comments

Point 1: just above section 2.4, the sentence that reads as follows: “After listening to each meow (participants could replay each sound ad libitum ) participants were asked to indicate the specific situation in which the vocalization was emitted (i.e. waiting for food, isolation, brushing), to indicate the emotional state of the meowing cat in terms of valence (positive vs. negative)...
It isn't entirely clear that the participants were given only those three alternatives (waiting for food, isolation, brushing) or whether they were free to offer other suggestions. I believe that those three alternatives were the only ones available to choose from, but because the sentence as it is written is ambiguous, I encourage a small revision/clarification there.

Response 1: We agree that this sentence was unclear and we have now specified this point as follows “ above. After listening to each meow (participants could replay each sound ad libitum), participants were asked to choose one out of three possible contexts in which the vocalization was emitted (i.e. waiting for food, isolation, brushing), to indicate the emotional state of the meowing cat in terms of valence (positive vs. negative), ( L251-254).

Point 2: in the discipline of sociolinguistics there is something called a "matched-guise test", which operates in a very similar way to the experimental technique described here. Because an important point of this paper is the investigation of attitudes towards the cat vocalizations that participants heard, I encourage the authors to include one or two references to that literature; if for no other reason to signal that the text is accessible to and may perhaps be useful to sociolinguists interested in human-animal communication.

Response 2: This is an interesting suggestion. The following sentence was added “ This procedure was based on previous research using qualitative behavior assessment in companion animals (dogs) [79] and reminds socio-linguistic research to investigate attitudes towards different aspects of language [80,81] (L 257-259)

Reviewer 3 Report

Authors present an interesting study about the human-cat relationship and how human try to understand cat feeling throughout sounds. Paper is well written and very easily to read for a broad audience.

I have only few comments to do.

Regarding social behaviour of cats (lines 53-56), there is a discussion about this statement. Some authors consider that cats, mainly domestic cats, are social but other commented that this complex behaviour has not been deeply studied based the scientific method, see for example explanations in Spotte (2014) Free-ranging cats. Behavior, Ecology, Management. Willey Blackwell. So, in this sentence a more general assumption could be done, such as "domestic cats show certain social interactions in particular circumstances" (for example around an abundant food resources...or something like this.

In line 47, scientific name of dogs and cats are not necessary here because they were used above.

Lastly, regarding questionnaire, I do not know if it is possible that the complete questionnaire is presented or available in a table or in an annex because it will be very useful to future similar studies in other places or situations with cats.

Congratulations to authors for this interesting contribution

Author Response

Response to Reviewer 3 Comments

Point 1 :  Regarding social behaviour of cats (lines 53-56), there is a discussion about this statement. Some authors consider that cats, mainly domestic cats, are social but other commented that this complex behaviour has not been deeply studied based the scientific method, see for example explanations in Spotte (2014) Free-ranging cats. Behavior, Ecology, Management. Willey Blackwell. So, in this sentence a more general assumption could be done, such as "domestic cats show certain social interactions in particular circumstances" (for example around an abundant food resources...or something like this.

Response 1: In line with  the reviewer’s suggestion we have modified the sentence  as follows: Differently from their wild ancestors (Felis silvestris), domestic cats are often defined to be social [32,33], as they show certain social interactions in particular circumstances (for example around an abundant food source), (L 53-55).

Point 2: Lastly, regarding questionnaire, I do not know if it is possible that the complete questionnaire is presented or available in a table or in an annex because it will be very useful to future similar studies in other places or situations with cats.

Response 2:  We have now added a link to the original questionnaire (available at https://www.lim.di.unimi.it/data/cat_survey/ 2020_survey.pdf, L-219-220). At present the questionnaire is in Italian, but we will update it with an English version as soon as possible.

Round 2

Reviewer 1 Report

Dear Authors,

As I wrote in my earlier review, I really enjoyed to read this manuscript, I think there are many valuable points and methodological strengths of it.

Thank you for responding my comments and criticisms. 

I wish this research would be continued in the future, and I truly recommend that you should broaden the sample size regarding the use of more numerous sound samples in the playback study.